# Skin Rejuvenation Efficacy and Safety Evaluation of *Kaempferia parviflora* Standardized Extract (BG100) in Human 3D Skin Models and Clinical Trial

**DOI:** 10.3390/biom14070776

**Published:** 2024-06-29

**Authors:** Wannita Klinngam, Phetploy Rungkamoltip, Ratjika Wongwanakul, Jaruwan Joothamongkhon, Sakkarin Du-a-man, Mattaka Khongkow, Udom Asawapirom, Tawin Iempridee, Uracha Ruktanonchai

**Affiliations:** National Nanotechnology Center (NANOTEC), National Science and Technology Development Agency, Pathum Thani 12120, Thailand; wannita.kli@nanotec.or.th (W.K.);

**Keywords:** *Kaempferia parviflora* standardized extract, polymethoxyflavones, anti-aging, 3D skin model, clinical trial

## Abstract

Polymethoxyflavones from *Kaempferia parviflora* rhizomes have been shown to effectively combat aging in skin cells and tissues by inhibiting senescence, reducing oxidative stress, and enhancing skin structure and function. This study assessed the anti-aging effects and safety of standardized *K. parviflora* extract (BG100), enriched with polymethoxyflavones including 5,7-dimethoxyflavone, 5,7,4′-trimethoxyflavone, 3,5,7,3′,4′-pentamethoxyflavone, 3,5,7-trimethoxyflavone, and 3,5,7,4′-tetramethoxyflavone. We evaluated BG100’s impact on skin rejuvenation and antioxidant properties using photoaged human 3D full-thickness skin models. The potential for skin irritation and sensitization was also assessed through studies on reconstructed human epidermis and clinical trials. Additionally, in vitro genotoxicity testing was performed following OECD guidelines. Results indicate that BG100 promotes collagen and hyaluronic acid production, reduces oxidative stress, and minimizes DNA damage in photoaged full-thickness 3D skin models. Furthermore, it exhibited non-irritating and non-sensitizing properties, as supported by tests on reconstructed human epidermis and clinical settings. BG100 also passed in vitro genotoxicity tests, adhering to OECD guidelines. These results underscore BG100′s potential as a highly effective and safe, natural anti-aging agent, suitable for inclusion in cosmeceutical and nutraceutical products aimed at promoting skin rejuvenation.

## 1. Introduction

Skin aging is a multifaceted process influenced by both intrinsic (e.g., age, hormones, and genetics) and extrinsic (e.g., sunlight and pollution) factors, resulting in noticeable alterations, such as wrinkles, sagging, and uneven pigmentation, as well as a decline in the skin’s barrier function and repair capacity [1]. Intrinsic or chronological factors contribute to skin thinning, a flattened dermal–epidermal junction, irregular pigmentation, collagen and elastin reduction, increased wrinkle formation, and decreased elasticity [2]. In contrast, extrinsic UV-induced photoaging manifests as thickened skin, prominent wrinkles, uneven pigmentation, amorphous elastic fibers, and damaged collagen in the dermis [1]. Emerging evidence indicates that skin aging is linked to cellular senescence, dysfunctional mitochondria, excessive reactive oxygen species (ROS), and oxidative stress in both the epidermis and dermis [3,4]. Natural compounds capable of mitigating skin aging have the potential to serve as effective bioactive agents in the creation of anti-aging cosmeceutical products.

*Kaempferia parviflora*, also known as Krachaidum in Thai, is a medicinal herb belonging to the Zingiberaceae family, native to northern and northeastern Thailand and also found in Laos and Malaysia [5,6]. *K. parviflora* is rich in polymethoxyflavones, including 5,7-dimethoxyflavone (DMF), 5,7,4′-trimethoxyflavone (TMF), 3,5,7,3′,4′-pentamethoxyflavone (PMF), 3,5,7-trimethoxyflavone (3TMF), and 3,5,7,4′-tetramethoxyflavone (TEMF) [5,7,8,9,10]. DMF has demonstrated anti-photoaging, anti-cancer, anti-cholinesterase, anti-inflammatory, and melanogenic activities [11,12,13,14,15]. TMF has displayed anti-inflammatory, anti-cholinesterase, lipolytic, and neuroprotective properties [7,16,17,18,19]. PMF is known for its lipolytic, neuroprotective, and antioxidant effects [7,18,19]. 3TMF has ameliorated skin damage by reducing proinflammatory cytokines and ROS [9]. Also, TEMF has exhibited anti-fungal and anti-mycobacterial activities [20]. Notably, *K. parviflora* extract (KPE) has been reported to inhibit cellular senescence and mitochondrial dysfunction in human dermal fibroblasts, alleviate aging symptoms in middle-aged and UVB-exposed hairless mice, and prolong the lifespan of *Caenorhabditis elegans* [21,22,23]. Additionally, we have previously studied the impact of DMF, TMF, and PMF on reducing cellular senescence and mitochondrial dysfunction in primary human dermal fibroblasts and revitalizing chronological aging in human ex vivo skin [24]. However, the commercialization of purified compounds is constrained by low yields and high costs. Utilizing *K. parviflora* as a plant extract enriched with polymethoxyflavones offers a more feasible strategy for incorporating these bioactive components into commercial products.

In this study, we aimed to evaluate the anti-aging properties of *K. parviflora*-derived plant extract, specifically BG100 extract, on UV-irradiated human full-thickness 3D skin tissue models that closely mimic natural skin structure and physiology. Our analysis included a comparison of BG100 extract’s efficacy in modulating collagen type I synthesis, hyaluronic acid secretion, intracellular ROS levels, 8-Hydroxy-2′-deoxyguanosine (8-OHdG) levels, and IL-6 secretion to that of well-known anti-aging agents, such as ethyl ascorbic acid and retinol. Additionally, we assessed the safety of BG100 extract by conducting in vitro skin irritation tests (OECD TG 439), in vitro genotoxicity evaluations (OECD TG 487), and skin irritation and sensitization tests with human volunteers. This research offers a comprehensive exploration of the anti-aging attributes and safety profiles of the BG100 extract, highlighting its potential as a valuable active component in anti-skin aging cosmeceutical and nutraceutical products.

## 2. Materials and Methods

### 2.1. Extraction and Isolation

Dried *Kaempferia parviflora* rhizomes were sourced from Phetchabun province, Thailand. One kilogram of the dried rhizomes was ground before being extracted with 10 L of ethyl acetate for six hours at 15 °C, stirred at a rate of 50 rpm. The resulting solution was filtered, concentrated under reduced pressure using a rotary evaporator, and lyophilized to yield the dry extract (BG100) at 4.03%. High-Performance Liquid Chromatography (HPLC) analysis of the dry extract was conducted using a reverse-phase HPLC system equipped with a photodiode array detector set to a wavelength of 254 nm (Agilent^®^ 1260 series, Agilent Technologies, Inc., Santa Clara, CA, USA) with an Agilent ZORBAX Eclipse XDB-C18 column (4.6 × 250 mm, 5 μm). Five standard compounds, DMF, TMF, PMF, 3TMF, and TEMF, were dissolved in methanol (HPLC grade, Fisher Chemicals, Loughborough, UK) and injected at a volume of 5 μL. The mobile phase system, comprising methanol and water, operated under the following gradient conditions: 65% methanol (flow rate: 0.6 mL/min) for 0–35 min, increased to 80% methanol for 35–40 min (flow rate: 0.6 mL/min), and maintained at 80% methanol for 40–60 min (flow rate: 0.7 mL/min) (adapted from Joothamongkhon et al. [25]). Quantitative analysis was carried out using HPLC chromatography, where the area under the peak for each of the five major compounds was measured. These values were then compared with the standard curve of known concentrations of the pure compounds for determination.

### 2.2. Human Full-Thickness 3D Skin Culture

EpiDerm full-thickness-400 hydrocortisone-free (EFT-400-HCF) skin tissues were procured from MatTek Corporation (Ashland, MA, USA), along with the accompanying EpiDerm maintenance medium without hydrocortisone (EFT-400-MM-HCF). On the day of receipt (day 0), the EFT-400-HCF skin tissues were placed in 6-well plates and conditioned in the maintenance medium at 37 °C and 5% CO_2_ overnight, following the MatTek protocol. On day 1, tissues were transferred into sterile Phosphate Buffered Saline (PBS, Thermo Fisher Scientific, Waltham, MA, USA) and exposed to 7 J/cm^2^ UVA and 50 mJ/cm^2^ UVB using a UV simulator (UVA cube 400, Honle UV Technology, Gilching, Germany). Subsequently, tissues were supplied with fresh medium and topically treated with 20 µL of either 0.1% or 0.2% BG100 extract in ethanol, 1% ethanol (negative control for BG100 extract, CAS no. 64-17-5, Sigma-Aldrich, Saint Louis, MO, USA), 0.5% retinol in DMSO (active ingredient benchmark, #R7632, Sigma-Aldrich, Saint Louis, MO, USA), 0.5% DMSO (negative control for retinol, #D8418, Sigma-Aldrich, Saint Louis, MO, USA), 2% ethyl ascorbic acid in water (active ingredient benchmark, #86404-04-8, Celmyon Co., Ltd., Noda, Japan), or distilled water (negative control for ethyl ascorbic acid). The treatment was applied for 7 h, with two skin tissues used per treatment group (N = 2 tissues/group). After treatment, tissues were thoroughly rinsed with PBS and incubated overnight. UV exposure was repeated daily for five days, and topical treatments were applied daily for six days. Culture mediums were collected on days 5 and 6, while skin tissues were harvested on day 6.

### 2.3. Collagen Type I Immunofluorescence Staining in Skin Tissues

Skin tissues were collected after topical treatment for six days and fixed in 4% paraformaldehyde (#12606, Cell Signaling Technology, Danvers, MA, USA) with 4% sucrose in PBS. Tissues were immersed in 30% sucrose in PBS overnight, embedded in Tissue-Tek^®^ O.C.T. compound (Sakura Finetek USA, Inc., Torrance, CA, USA), and cryo-sectioned to 5 µm thickness using a Leica CM1860 (Leica Biosystems, Deer Park, IL, USA), and mounted on glass microslides at −20 °C. After washing with PBS, permeabilization with 0.1% Triton X-100 (#X100, Sigma-Aldrich, Saint Louis, MO, USA), and blocking with 1% bovine serum albumin (#A3294, Sigma-Aldrich, Saint Louis, MO, USA) in PBS, sections were incubated with anti-collagen I antibody (#ab34710, Abcam, Cambridge, MA, USA) at 4 °C overnight, followed by incubation with goat anti-rabbit IgG H&L Dylight^®^ 488 preabsorbed (#ab96899, Abcam, Cambridge, MA, USA) and 1:2000 DAPI (#62248, Thermo Fisher Scientific, Waltham, MA, USA). Slides were mounted with Prolong Gold Antifade mounting medium (#9071S, Cell Signaling Technology, Danvers, MA, USA) and left for 24 h. Collagen type I was visualized using a TCS SP8 confocal microscope (Leica Biosystems, Deer Park, IL, USA). Fluorescent intensity was quantified using ImageJ software version 1.54 (National Institutes of Health, Bethesda, MD, USA) and reported as integrated density. Three sections per tissue were imaged (N = 6 sections/group), with 6–8 images per section averaged for integrated density.

### 2.4. Hyaluronic Acid and IL-6 Secretion in Tissue Culture Mediums

Tissue culture mediums collected after six days of topical treatment were centrifuged at 2000× *g* and 4 °C for 10 min using a high-speed refrigerated microcentrifuge MX-305 (Tomy Seiko Co., Ltd., Tokyo, Japan). Supernatants were collected, and the expressions of hyaluronic acid (#AL254HVc, PerkinElmer, Waltham, MA, USA) and IL-6 (#AL223c, PerkinElmer, Waltham, MA, USA) were measured according to the manufacturer’s protocol. Data were recorded using an EnVision Microplate Reader (PerkinElmer, Waltham, MA, USA) and normalized to the total protein concentration, which was measured using the Pierce™ BCA Protein Assay Kit (#23225, Thermo Fisher Scientific, Waltham, MA, USA). Assays were performed in duplicate from two tissues per treatment.

### 2.5. ROS and 8-OHdG Levels in Tissue Culture Mediums

Tissue culture mediums collected after five days of UV and topical treatment were centrifuged at 10,000 RCF and 4 °C for 5 min using a high-speed refrigerated microcentrifuge MX-305 (Tomy Seiko Co., Ltd., Tokyo, Japan). Supernatants were collected, and ROS and 8-OHdG levels were measured using the DCF ROS/RNS Assay Kit (#ab238535, Abcam, Cambridge, MA, USA) and 8-hydroxy 2-deoxyguanosine ELISA Kit (#ab201734, Abcam, Cambridge, MA, USA), respectively. The manufacturer’s protocol for each assay kit was followed. Fluorescent intensity and absorbance were recorded on a SpectraMax M2 microplate reader (Molecular Devices, LLC., San Jose, CA, USA) and normalized to the total protein concentration, which was measured using the Pierce™ BCA Protein Assay Kit (#23225, Thermo Fisher Scientific, Waltham, MA, USA). Assays were performed in duplicate from two tissues per treatment.

### 2.6. In Vitro Skin Irritation Tests Using Reconstructed Human Epidermis Skin (OECD Guidelines No. 439)

An MTT interference test was performed to avoid potential interference by colored testing articles. Briefly, 0.2% BG100 extract, PBS (negative control), or potassium hydroxide (positive control) was incubated with MTT reagent at 37 °C and 5% CO_2_ for 3 h. Color changes were observed to identify interference.

For in vitro skin irritation tests, EPI-200 skins (MatTek Corporation) were incubated in assay medium at 37 °C and 5% CO_2_ overnight. Tissues were exposed to 0.1% and 0.2% BG100 extracts, PBS (negative control), or 5% SDS (positive control) for 1 h (N = 3 tissues/group). After a 24-h incubation, IL-1α expression and tissue viability were assessed. The human IL-1α AlphaLISA Detection Kit (#AL238C, PerkinElmer) and MTT solution were used for these analyses, with data normalized to total protein concentration. If the mean percent tissue viability after exposure is <50%, the testing article is considered an irritant [26].

### 2.7. In Vitro Mammalian Cell Micronucleus Test (OECD Guidelines No. 487)

We assessed BG100 extract’s genotoxic effects on V79-4 cells (CCL-93TM, ATCC, Manassas, VA, USA), a male Chinese hamster lung, using the cytokinesis-blocked micronucleus (CMBN) method. The test was conducted for 4 h (short term) with and without metabolic activation, and for 24 h (long term) without metabolic activation. Cells in 6-well plates were grown in DMEM (#11995-065, Gibco, Waltham, MA, USA) with 10% FBS (#10270, Gibco, MA, USA) and 1% penicillin–streptomycin (#15140, Gibco, Waltham, MA, USA). They were treated with BG100 extract, positive controls (mitomycin C (MMC, CAS no. 50-07-7, HiMedia, Mumbai, India) or benzo[a]pyrene (CAS no. 50-32-8, Sigma-Aldrich, Saint Louis, MA, USA), or culture medium (negative control) for 4 or 24 h. Cytochalasin B (CAS no. 14930-96-2, Sigma-Aldrich, Saint Louis, MA, USA) was added to experiments accordingly, and cells were incubated at 37 °C and 5% CO_2_.

Before assessing genotoxicity, the cytotoxicity of BG100 extract was examined using MTT assays. V79-4 cells in 96-well plates were cultured for 24 h, followed by treatment with concentrations of 50, 100, 200, and 400 µg/mL, or a negative control (culture medium) for 4 and 24 h. Cell viability was measured by evaluating metabolic activity with the MTT dye (CAS No. 298-93-1, Sigma-Aldrich, Saint Louis, MA, USA). A cytotoxic effect was considered significant if there was a reduction in cell viability greater than 30%.

For metabolic activation, 2% *v*/*v* of S9 fraction (#RTS9PL, Gibco, Waltham, MA, USA) was added to each well. After 4-h treatment and cytochalasin B addition, cells were incubated at 37 °C and 5% CO_2_. Cells were then removed using 0.25% trypsin-EDTA (#25200, Gibco, Waltham, MA, USA), washed with DPBS, and added to 0.56% *w*/*v* KCl. After centrifugation, the cell pellets were resuspended in fixative solution I, which consisted of a mixture of 0.9% NaCl solution (CAS no. 7647-14-5, Merck, Søborg, Denmark), methanol (CAS no. 67-56-1, LabScan, Bangkok, Thailand), and acetic acid (CAS no. 64-19-7, Merck, Søborg, Denmark). The cells were then fixed twice with fixative solution II, which contained methanol and acetic acid. The fixed cell suspension was stored at 4 °C until further use.

Glass slides were soaked in methanol, immersed in fixative solution II, and dried with V79-4 cell suspension. After staining with Fluoroshield^TM^ with DAPI (CAS no. 28718-90-3, Sigma-Aldrich, Saint Louis, MA, USA), micronucleus (MN) frequency and cytokinesis-block proliferation index (CBPI) were scored using Metafer scanning and imaging platform (Metafer^®^, Axio Imager.Z2, Zeiss, Oberkochen, Germany) and manually counterchecked [27]. The cytokinesis-blocked binucleated cells were scored for MN frequency with specific characteristics as described [28]. All experiments were performed in triplicate, with MN frequencies and CBPI values expressed as mean and standard error of the mean (SEM).

### 2.8. Human Volunteer Skin Irritation and Sensitization Tests

The clinical skin irritation and sensitization test was carried out by Dermscan Asia Co., Ltd. The sensitizing potential of 0.2% BG100 extract was evaluated in 52 healthy volunteers (47 females and 5 males, aged 23–60 years old), following the inclusion and non-inclusion criteria [29]. The study received approval from the Institutional Ethics Committee (EXP#1231) and was conducted in compliance with the Declaration of Helsinki. All participants provided informed consent and adhered to specific guidelines throughout the study.

Finn Chamber^®^ 8 mm occlusive patches (Bio-diagnostics, Worcester, UK) containing 0.2% BG100 extract or butylene glycol (negative control) were applied to the homolateral-side scapular zones. Patch tests were conducted on Monday, Wednesday, and Friday for three consecutive weeks, with evaluations after 48 h on weekdays and 72 h on weekends as shown in Appendix A. Erythema and edema were scored by the investigator, following the criteria and scale detailed in Table 1.

Skin irritation testing involved calculating the Cumulative Irritation Index (C.I.I.) for each participant by dividing the sum of erythema and edema scores by the number of readings. The Mean Cumulative Irritation Index (M.C.I.I.) was calculated by dividing the sum of the Cumulative Irritation Index (C.I.I.) by the number of subjects. However, if an irritation reaction was observed in the negative control (butylene glycol), the reading was disregarded. The irritation potential was classified based on M.C.I.I. values according to Table 2.

After 14 days of the last induction patch, a challenge patch was applied to the opposite side for 48 h, and allergic reactions were assessed using the International Contact Dermatitis Research Group (ICDRG) criteria (Table 3) after 30 min and 48 h of patch removal. A score of 2 or higher was indicative of a contact allergy reaction, and if at least one case of active sensitization was present, the product was defined as “potentially sensitizing”.

### 2.9. Statistical Analysis

For inter-group comparisons, unpaired Student’s *t*-test and one-way ANOVA with Tukey’s post-hoc test were employed for two and multiple groups, respectively. Statistical analyses were conducted using GraphPad Prism 9 (GraphPad Software, San Diego, CA, USA). The significance level was set at *p* < 0.05, and the results are presented as the mean ± standard deviation unless otherwise specified.

## 3. Results

### 3.1. Polymethoxyflavone Composition in BG100 Extract

Standard polymethoxyflavones were utilized for the peak identification of the HPLC fingerprint. Appendix A illustrates the HPLC chromatogram of five standard methoxyflavones, DMF, TMF, PMF, 3TMF, and TEMF, while Appendix A presents the chromatogram of the BG100 extract. The structures of these five standard compounds are depicted in Appendix A. Quantitative HPLC analysis revealed that 1 g of dried BG100 extract contained 157.60 mg of PMF, 166.55 mg of DMF, 172.78 mg of TMF, 42.24 mg of 3TMF, and 80.53 mg of TEMF, totaling 619.70 mg of polymethoxyflavones, constituting 61.97% of the BG100 extract.

### 3.2. BG100 Extract-Induced Collagen Type I Stimulation in UV-Exposed Human Full-Thickness 3D Skin Tissues

Long-term UV exposure in human skin leads to collagen damage and the disorganization of dermal collagen fiber bundles, which are critical characteristics of photoaging [30,31]. UV radiation triggers the production of matrix metalloproteinase (MMP)-1, which cleaves collagen fibers, specifically types I and III, compromising the structural integrity of the dermis [32]. Additionally, UV irradiation reduces procollagen types I and III, associated with newly synthesized collagen in human skin [33].

To determine whether the BG100 extract, enriched with polymethoxyflavones, could restore collagen type I expression in UV-exposed human skin tissues, we performed collagen type I immunofluorescence staining in full-thickness 3D skin models. As shown in Figure 1a,b, UV irradiation decreases collagen type I expression by 48.69–52.80% compared to non-UV-exposed tissues. Conversely, 0.5% retinol increases collagen type I expression by 51.17% compared to DMSO-UV-exposed tissues, while 2% ethyl ascorbic acid has no significant effect. Both 0.1% and 0.2% BG100 extract increase collagen type I expression by 37.83% and 35.13%, respectively, compared to ethanol-UV-exposed tissues. These findings suggest that the polymethoxyflavone-enriched BG100 extract effectively stimulates collagen type I production in UV-exposed human 3D skin tissues.

### 3.3. BG100 Extract-Stimulated Hyaluronic Acid Secretion in UV-Exposed Human Full-Thickness 3D Skin Tissues

Hyaluronic acid, a component of the extracellular matrix, plays a crucial role in skin hydration, aging, and wound healing [34]. Previous studies have shown that hyaluronic acid levels decrease in response to both acute and chronic UV irradiation in human and mouse skin tissues [35]. In the present study, we explored the potential of BG100 extract to stimulate hyaluronic acid secretion in the culture medium of UV-exposed human skin tissues. As illustrated in Figure 2, UV exposure led to a reduction in hyaluronic acid secretion by 60.33–85.01% compared to non-UV-treated skin tissues. The 2% ethyl ascorbic acid and 0.5% retinol enhanced hyaluronic acid secretion by 39.52% and 299.22%, respectively. Additionally, 0.1% and 0.2% BG100 extracts increased hyaluronic acid secretion by 73.43% and 46.01%, respectively, compared to ethanol-UV-exposed tissues. Consequently, our findings indicate that BG100 extract effectively restores hyaluronic acid synthesis in UV-exposed human 3D skin tissues.

Additionally, UV radiation activates several signaling pathways associated with inflammatory cytokines [36]. Previous studies have found that UV activates the transcription factor NF-κB, which can increase proinflammatory cytokine genes, such as IL-6 [32]. Consequently, we investigated the effect of BG100 extract on the inhibition of IL-6 secretion in culture mediums. As shown in Appendix A, UV exposure increases IL-6 secretion by 117.2–316.93% compared to non-UV-treated tissues. Both 2% ethyl ascorbic acid and 0.5% retinol decrease IL-6 secretion by 19.17% and 27.92%, respectively, compared to the UV-exposed negative control. However, neither 0.1% nor 0.2% BG100 extract reduces IL-6 expression in culture mediums of human 3D skin tissues. This suggests that BG100 extract might not influence the reduction in IL-6 expression following UV treatment. The effects of BG100 extract on other inflammatory and senescence-associated secretory phenotype (SASP) markers will be explored in future studies.

### 3.4. BG100 Extract-Mediated ROS Inhibition in UV-Exposed Human Full-Thickness 3D Skin Tissues

Oxidative stress is a critical cellular response following UV irradiation of the skin. UV exposure can increase hydrogen peroxide levels and other ROSs, damaging various cellular components, including DNA, proteins, and lipids [32,37,38]. To evaluate the capacity of BG100 extract in mitigating oxidative stress in photoaged skin, we analyzed the release of reactive oxygen species (ROS) in the cell culture media. As illustrated in Figure 3, UV exposure induced a substantial increase in ROS levels by 40.32–71.19% compared to non-UV-exposed skin tissues. Treatment with 0.5% retinol led to a notable 28.54% reduction in ROS levels in UV-exposed tissues, while 2% ethyl ascorbic acid demonstrated no significant effect. Notably, both 0.1% and 0.2% BG100 extracts exhibited a decrease in ROS release by 32.84% and 35.25%, respectively, in comparison to ethanol-UV-exposed tissues. These findings underscore the antioxidant properties of BG100 extract, evident in the reduction in ROS levels in the culture media of UV-exposed human 3D skin tissues.

### 3.5. BG100 Extract-Mediated Attenuation of DNA Damage in UV-Exposed Human Full-Thickness 3D Skin Tissues

ROS can damage various cellular components, including DNA, leading to mutagenesis, carcinogenesis, and aging [36]. One key biomarker for oxidative DNA damage is 8-hydroxy-2′-deoxyguanosine (8-OHdG), which increases with skin photoaging [39]. To assess the potential of BG100 extract in ameliorating UV-induced DNA damage, we quantified 8-OHdG levels in the tissue culture media. As depicted in Figure 4, UV exposure leads to a significant increase in 8-OHdG levels by 94.30–102.92% compared to non-UV-exposed tissues. Treatment with 0.5% retinol demonstrates a substantial 42.37% reduction in 8-OHdG levels when compared to DMSO-UV-exposed tissues, while ethyl ascorbic acid exhibits no significant effect. Furthermore, both 0.1% and 0.2% BG100 extracts display inhibitory effects on 8-OHdG release, showing reductions of 32.42% and 44.75%, respectively, compared to ethanol-UV-exposed tissues. In summary, these findings indicate that BG100 extract confers a protective effect against DNA damage induced by UV exposure in human 3D skin tissues.

### 3.6. Absence of Irritation by 0.2% BG100 Extract on In Vitro Reconstructed 3D Human Skin Tissues

To assess the safety of BG100 extract for skin application, we conducted an in vitro skin irritation test on reconstructed human epidermis skin according to the OECD Test Guideline 439. We used the MTT tissue viability assay to evaluate the potential skin irritation caused by the extract [40]. Before the test, we performed an MTT interference assay to confirm that the extract did not alter the color of the MTT reagent and produce false-positive results. As shown in Figure 5a, while the positive control (KOH) changed the color of the MTT solution from yellow to purple, the 0.2% BG100 extract did not. Based on the MTT interference assay, the tested concentration of BG100 extract did not interfere with the MTT solution. Hence, the in vitro skin irritation test can be conducted without the need for frozen tissue subtraction [26].

As per the guidelines of OECD TG 439, a decrease in cell viability exceeding 50% is an indication of skin irritation. The positive control (5% SDS) caused a reduction in cell viability below 50%, while the 0.2% BG100 extract did not show any adverse effects on cell viability, suggesting that the extract did not cause skin irritation (Figure 5b). Additionally, the release of IL-1α protein can serve as the secondary endpoint of the skin irritation test according to the European Center for the Validation of Alternative Methods (ECVAM) [41]. A skin irritant is defined when the testing article induces IL-1α expression three-times higher than the negative control [42]. While 5% SDS robustly induced IL-1α secretion above the threshold, the BG100 extract did not, confirming that BG100 extract is non-irritant to the skin (Figure 5c).

### 3.7. Non-Genotoxicity of BG100 Extract on V79-4 Cells

The induction of micronuclei (MN) is a well-established cytogenetic endpoint for assessing structural and numerical chromosomal alterations in genotoxicity testing [43]. MN are small extranuclear bodies found in interphase cells resulting from chromosome breaks and lagging chromosomes that fail to incorporate into the main nucleus during cell division [28]. Measuring MN frequency can be used to screen drug candidates and other test chemicals for genotoxicity. To evaluate the potential genotoxicity of BG100 extract on mammalian cells (V79-4 cells), we used the CBMN method in accordance with OECD guidelines for testing of chemicals no. 487 [27,28].

We first conducted an MTT assay to evaluate the cell viability of V79-4 cells and determine the optimal concentration for genotoxicity testing. According to established guidelines, the highest concentration of the test substance should aim to achieve 55 ± 5% cytotoxicity in the target cells. Results from Appendix A show that treatment with 50, 100, 200, and 400 µg/mL of BG100 extract for 4 h only marginally affects cell viability. However, exposure to 400 µg/mL of BG100 extract for 24 h reduced the cell survival rate to 45.3%, indicating that this should be the maximum concentration used for genotoxicity evaluation.

To ensure the accuracy of the results, we also evaluated cytotoxicity using the cytokinesis-block proliferation index (CBPI) since some testing substances may cause cytotoxicity through the inhibition of cell division and affect cell proliferation and cell scoring. The CBPI is calculated from the number of micronucleus cells (MNCs) and binucleate cells (BNCs) within the same area on a slide containing at least 500 cells per culture [27]. As shown in Figure 6b, all concentrations of BG100 extract did not inhibit cell division, as indicated by CBPI values greater than 1 [27].

Next, we determined micronucleus frequency by counting the number of micronucleus cells (MNCs) per 2000 binucleate cells (BNCs) using Metafer^®^ after staining the cells with DAPI [27]. In comparison to the negative control (culture medium), cells treated with 1.25 µg/mL MMC (positive control) showed a significant increase in MN frequencies. However, no significant difference was observed in MN frequencies in cells treated with BG100 extract at concentrations of 50, 100, 200, or 400 µg/mL for either 4 or 24 h in the absence of S9 fraction (Figure 6a). These results suggest that the tested concentrations of BG100 extract do not exhibit genotoxicity toward V79-4 cells.

To assess the potential genotoxicity of BG100 extract when combined with a metabolic activator, we performed experiments incorporating the S9 fraction during 4-h treatments. Figure 7a illustrated that cells treated with 5 µg/mL benzo[a]pyrene (BP; positive control) exhibited a significant increase in MN frequencies compared to the negative control. Conversely, cells treated with BG100 extract at concentrations of 50, 100, 200, or 400 µg/mL for 4 h in the presence of S9 fraction showed no significant differences in MN frequencies relative to the negative control. Furthermore, the cytokinesis-block proliferation index (CBPI) was calculated, and values greater than 1 indicated that none of the tested concentrations of BG100 extract inhibited cell division (Figure 7b). Based on these findings, we conclude that BG100 extract is not genotoxic in the presence of a metabolic activator under the experimental conditions employed.

### 3.8. Non-Irritating and Non-Sensitizing Properties of 0.2% BG100 Extract, as Demonstrated in A Clinical Trial

We next investigated the safety of BG100 extract in clinical testing with a focus on its skin sensitizing and irritation potential. Figure 8 shows the flowchart illustrating the enrollment process, topical intervention, follow-up procedures, and analysis for the groups. The 0.2% BG100 extract was assessed in 54 healthy subjects, who were initially selected based on inclusion and exclusion criteria [29]. However, two participants withdrew from the study, resulting in a final cohort of 52 individuals (47 females and 5 males). Participant characteristics, such as sex, age, phototype, and medical and surgical treatments prior to and during the study, are detailed in Appendix A.

During the induction phase of our study, we applied 0.2% BG100 extract, mixed with butylene glycol, or butylene glycol alone (negative control), to the backs of each subject every other day for three consecutive weeks. After 48 h of application on weekdays and 72 h on weekends, we scored the irritation potential related to erythema and edema at the topical sites. This process was repeated nine times. As shown in Appendix A, the results reveal minimal or no erythema and edema in most subjects during the induction phase. The Mean Cumulative Irritation Index (M.C.I.I.) was calculated using the Cumulative Irritation Index (C.I.I.), representing the average erythema and edema scores for each participant. According to the classification outlined in Table 2, our findings indicate that the 0.2% BG100 extract, with an M.C.I.I. of 0.08, is characterized as non-irritating to the skin of human volunteers.

Following a 2-week resting period, a challenge patch was applied to the naive site for 48 h, and skin allergic reactions were evaluated after 30 min and 48 h of patch removal. The results, presented in Table 4, indicate that there are no subjects with allergic reaction scores of ++ (2) or +++ (3), indicating that the 0.2% BG100 extract is classified as non-sensitizing to human skin. Therefore, we conclude that 0.2% BG100 extract is safe for use on human skin and can be classified as “Non-irritating” and “Non-sensitizing”.

## 4. Discussion

The skin undergoes aging through two primary mechanisms: internally, due to the chronological aging process, and externally, as a result of environmental factors, particularly UV irradiation, which can lead to premature aging [32]. Despite the distinct appearances of photoaged and chronologically aged human skin, both aging processes share several key molecular features. These include the promotion of matrix metalloproteinase (MMP) expression, a decrease in procollagen synthesis and collagen expression, damage to extracellular matrix components, generation of reactive oxygen species (ROS), and activation of proinflammatory cell signaling pathways [32,35,36,39].

Both UVA and UVB contribute significantly to photoaging. UVB primarily affects the epidermis, where it is largely absorbed, and induces pyrimidine dimer-type DNA damage through direct absorption [44]. Chronic UVB exposure has been shown to downregulate hyaluronic acid synthase in the mouse dermis layer by inhibiting the transforming growth factor (TGF)-β signaling pathway [45]. Moreover, low doses of UVB have been found to upregulate transcription factors, such as activator protein 1 (AP-1) and nuclear factor-kappa B (NF-κB), which in turn lead to increased cytokine production and matrix metalloproteinase (MMP) activity in human skin [30]. UVA has the ability to penetrate deeper into the dermis compared to UVB and has been associated with damage to dermal connective tissues, oxidative stress induction, and skin inflammation via the AP-1/NF-κB pathway [46,47,48]. In addition, UVA indirectly causes base oxidation through UV-induced ROS production [44]. Consequently, to study the anti-aging effects of BG100 extract, it is essential to consider both UVA and UVB exposure, as they jointly contribute to photoaging skin.

A human 3D full-thickness skin model is composed of normal human epidermal keratinocytes and human dermal fibroblasts, which form a multilayered dermis and epidermis. The skin is cultured at the air–liquid interface, achieving a differentiation state similar to native skin, as demonstrated by the expression of keratin 5, keratin 10, and involucrin [49]. The 3D full-thickness skin model features organized epidermal layers, which play a crucial role in skin absorption. Additionally, epidermal and dermal interactions significantly affect the secretion of inflammatory mediators [50]. A prior study has demonstrated that epidermal skin equivalents lack the ability to release cytokines such as IL-6. In contrast, human 3D full-thickness skin models, which include dermal fibroblasts, can release several inflammatory markers [51]. These findings suggest that a human 3D full-thickness skin model closely resembles human skin, allowing for the evaluation of cosmetic ingredients and products under in vivo-like conditions. Owing to its close similarity to authentic human skin, we selected a human 3D full-thickness skin model to study the anti-aging effects of BG100 extract after UV exposure.

In our previous study, we demonstrated the anti-aging effects of DMF, primarily TMF and PMF, in primary human dermal fibroblasts and chronologically aged ex vivo skin models [24]. In the present study, we developed a standardized BG100 extract by enriching the yield of TMF and PMF from 20% to 33%, aiming to enhance its potential anti-aging effects on photoaged skin tissues. Our findings show that BG100 extract stimulates collagen type I and hyaluronic acid secretion while inhibiting ROS and DNA damage in UV-induced human 3D full-thickness skin (Figure 1, Figure 2, Figure 3 and Figure 4). These results align with the known biological effects of these compounds.

Additionally, BG100 extract is enriched with 3TMF, which has been reported to exhibit antioxidant and anti-inflammatory properties by reducing ROS, matrix metalloproteinase (MMP)-1, IL-1β, IL-6, IL-8, and cyclooxygenase-2 (COX-2) induced by tumor necrosis factor (TNF)-α in normal human fibroblasts [9]. As a result, we hypothesize that the increased presence of 3TMF in BG100 extract may contribute to greater anti-aging benefits.

In addition to the efficacy of BG100 extract in mitigating skin aging, we have also thoroughly assessed the safety of the extract for the skin. We initially evaluated the skin irritation potential of a 0.2% BG100 extract using reconstructed human epidermis, following the OECD Test Guideline No. 439. Chemical-induced skin irritation commences when chemicals penetrate the stratum corneum, causing damage to keratinocytes and other skin cells [50]. The damaged cells initiate inflammatory cascades, signaling cells in the dermis, particularly stromal and endothelial cells, resulting in erythema and edema [50]. Therefore, employing reconstructed human epidermis, consisting of epidermal keratinocytes that form a multi-layered stratum corneum resembling in vivo skin, is a valuable method for evaluating skin irritation [26]. Our study demonstrates that BG100 extract does not cause skin irritation in reconstructed human epidermis, as evidenced by MTT viability and IL-1α expression assays (Figure 5). This finding is consistent with the cumulative irritation test results observed in human volunteers after three consecutive weeks of BG100 extract application (Appendix A).

We further assessed the genotoxicity of BG100 extract through an in vitro mammalian cell micronucleus test, following OECD Test Guideline No. 487. This test detects micronuclei induced by aneugens (substances causing structural chromosomal aberrations through DNA breakage) or clastogens (substances leading to numerical chromosomal aberrations by interacting with targets other than DNA) in the cytoplasm of interphase cells [52,53]. The procedure includes short-term treatment both with and without metabolic activation (S9 fraction) and long-term treatment without it [27]. By adding cytochalasin B, an actin polymerization inhibitor, before cell mitosis, only binucleate cells that have completed one mitosis are analyzed for genotoxicity [54]. The evaluation of micronucleus induction is based on the frequencies of binucleate cells with micronuclei, and cytostasis should be determined [55]. Our study found that BG100 extract, at concentrations of 50–400 µg/mL, did not exhibit genotoxicity in mammalian cells, regardless of the presence (4-h exposure) or absence (4- and 24-h exposure) of metabolic activation (Figure 6 and Figure 7).

Moreover, we assessed the skin sensitization potential of BG100 extract in human volunteers using the Human Repeated Insult Patch Test (HRIPT). This protocol helps determine whether repeated application of BG100 extract under maximized conditions can induce allergic contact dermatitis. The HRIPT protocol consists of two phases: the induction phase and the challenge phase. During the induction phase, a patch is applied for three consecutive weeks with nine total applications. After a two-week rest period, the challenge phase involves a single 48-h patch application of BG100 extract to a new, unexposed site. Skin reactions observed at the new site during the challenge phase can indicate dermal sensitization [56]. Our results show that 0.2% BG100 extract does not cause erythema, edema, vesicles, or post-blister ulcerations in human volunteers, suggesting that 0.2% BG100 extract is non-sensitizing and does not induce allergic contact dermatitis in human skin (Table 4 and Appendix A).

In summary, our study demonstrates the potential of the standardized BG100 extract from *K. parviflora* in promoting skin rejuvenation and providing antioxidant protection for photoaged human 3D full-thickness skin models. At the same time, our findings indicate that BG100 extract is safe, as evidenced by its non-irritating effect on human skin, lack of in vitro genotoxicity, and absence of skin sensitization. These results highlight the promising potential of BG100 extract as a safe and effective ingredient for anti-aging skincare formulations and nutraceuticals targeting skin health and rejuvenation. Further investigations are warranted to validate the clinical efficacy of BG100 extract in human subjects.

## 5. Conclusions

Our study demonstrates that the standardized BG100 extract, enriched with polymethoxyflavones isolated from *Kaempferia parviflora*, exhibits significant skin rejuvenation effects. It enhances collagen type I and hyaluronic acid secretion while reducing ROS and DNA damage in photoaged human 3D full-thickness skin models. Furthermore, BG100 extract has proven to be non-genotoxic in vitro, and non-irritating and non-sensitizing to human skin in both in vitro and clinical trials. Consequently, BG100 emerges as a promising and safe anti-aging active ingredient, ideal for integration into cosmeceutical and potentially nutraceutical products.

For the development of cosmeceutical products, we propose conducting future efficacy testing through clinical trials, specifically employing split-face experiments with a sample size of 30 participants. These trials will compare the efficacy of a standard base formulation against the same formulation enriched with BG100, assessing metrics such as hydration, elasticity, skin brightening, and wrinkle reduction. Volunteers selected for these trials will be those regularly exposed to sunlight, aligning clinical outcomes with data previously gathered from 3D skin model studies.

For the potential use of BG100 as an active ingredient in nutraceutical products, ongoing safety and efficacy testing in animal studies is essential. Our collaborative research teams have completed acute and subchronic toxicity studies to ensure safety. Current studies include experiments on mice treated with D-galactose to simulate accelerated aging, aiming to mitigate aging-related damage in various organs and tissues, including skin. These studies are crucial for evaluating improvements in tissue functions, reductions in oxidative stress, and decreases in cellular senescence markers, providing essential data for further clinical validation of BG100 for anti-aging and skin rejuvenation.

Ultimately, we envision that the successful development and commercialization of BG100 will position Thailand as one of the leaders in the global market for natural active ingredients. This initiative promises to significantly boost Thailand’s economic growth, particularly enhancing the livelihoods and economic stability of its rural communities, showcasing Thailand’s innovation in biotechnology, and promoting sustainable development in its agricultural sector.

## Figures and Tables

**Figure 1 biomolecules-14-00776-f001:**
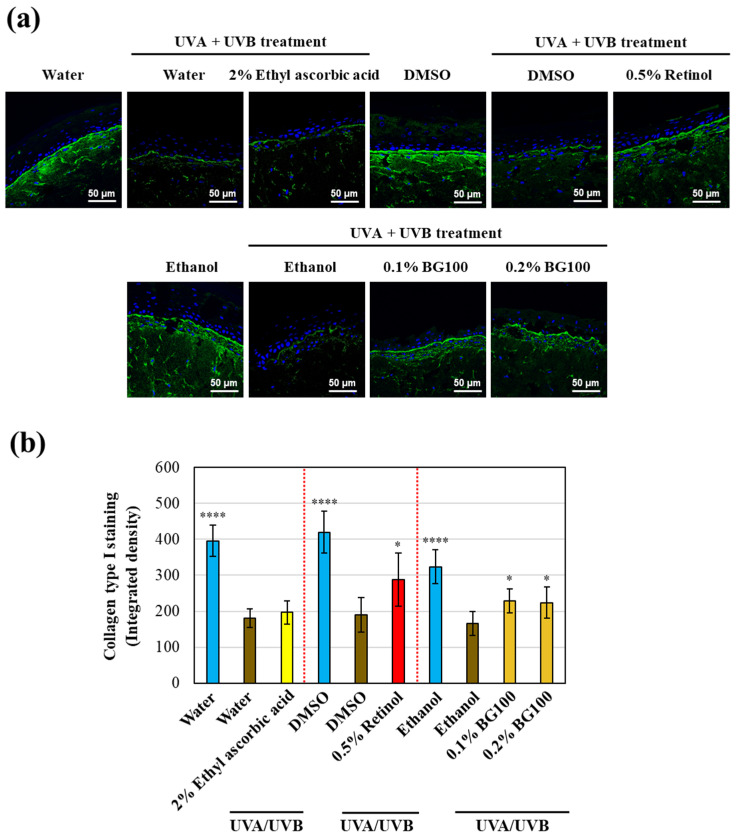
BG100 extract enhances collagen type I expression in UV-exposed human full-thickness 3D skin tissues. (**a**) Representative images of 3D skin tissues exposed or not exposed to UVA and UVB for 5 days, followed by topical application of 2% ethyl ascorbic acid in water, water, 0.5% retinol in DMSO, 0.5% DMSO, 0.1% and 0.2% BG100 extracts in ethanol, or 1% ethanol for 6 days, depicting collagen type I (green) and DAPI nuclear staining (blue). (**b**) Quantification of collagen type I intensity (**a**) expressed as integrated density. Two skin tissues were employed for each treatment, and 3 sections per tissue were imaged (6–8 images/section). * *p* < 0.05; **** *p* < 0.0001 compared to the UV-exposed negative control.

**Figure 2 biomolecules-14-00776-f002:**
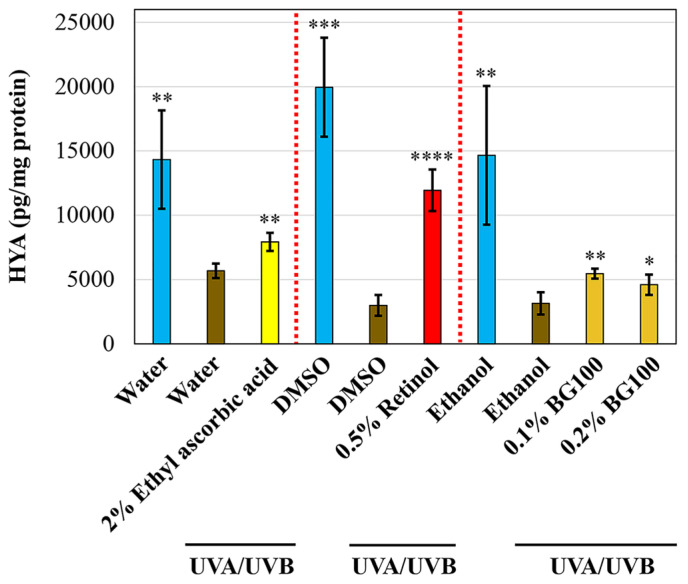
BG100 extract enhances hyaluronic acid secretion in UV-exposed human full-thickness 3D skin tissues. Hyaluronic acid secretion was measured in the culture medium of human full-thickness 3D skin tissues, which were either exposed or not exposed to UVA and UVB radiation for 5 days, and topically treated with 2% ethyl ascorbic acid in water, water, 0.5% retinol in DMSO, 0.5% DMSO, 0.1% and 0.2% BG100 extracts in ethanol, or 1% ethanol, for 6 days. Data are based on duplicate measurements from two skin tissues for each treatment. * *p* < 0.05; ** *p* < 0.01; *** *p* < 0.001; **** *p* < 0.0001, compared to the UV-exposed negative control.

**Figure 3 biomolecules-14-00776-f003:**
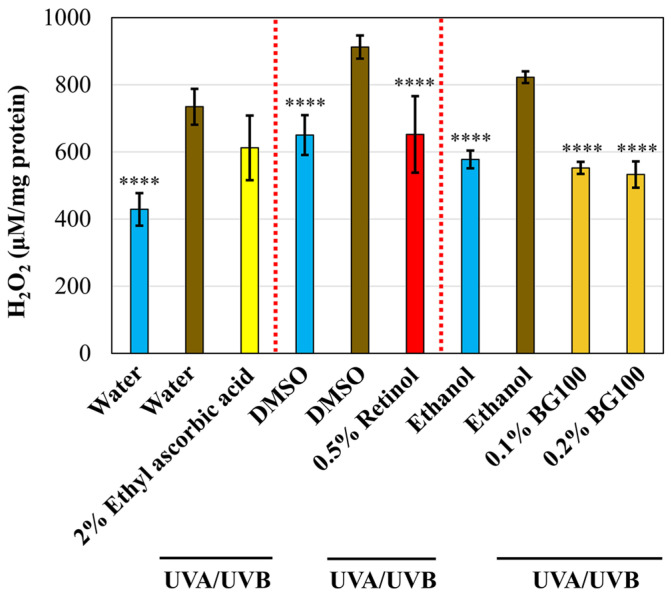
The antioxidant effect of BG100 extract demonstrated by inhibiting Reactive Oxygen Species (ROS) in UV-exposed human full-thickness 3D skin tissues. ROS level was measured in the culture media of human full-thickness 3D skin tissues that were exposed or not exposed to UVA and UVB for 5 days and topically treated with 2% ethyl ascorbic acid in water, water, 0.5% retinol in DMSO, 0.5% DMSO, 0.1% and 0.2% BG100 extracts in ethanol, or 1% ethanol for 6 days. Data are based on duplicate measurements from two skin tissues per treatment. **** *p* < 0.0001, compared to the UV-exposed negative control.

**Figure 4 biomolecules-14-00776-f004:**
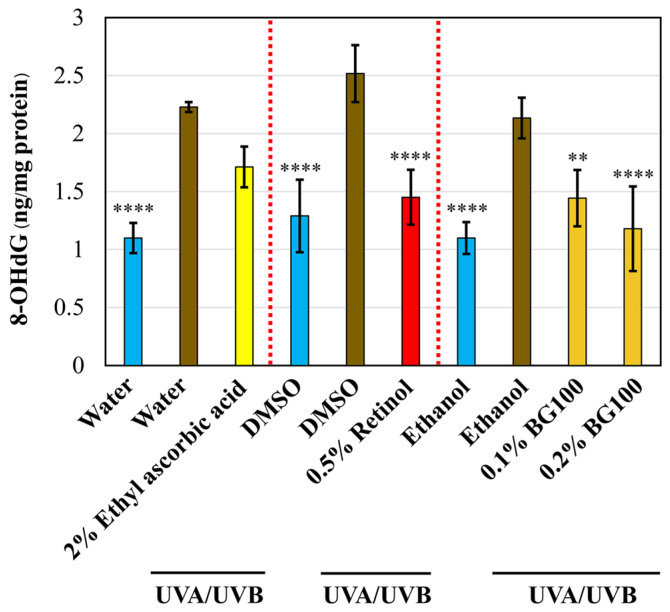
BG100 extract protects against DNA damage in UV-exposed human full-thickness 3D skin tissues. 8-Oxo-2′-deoxyguanosine (8-OHdG) was measured in the culture medium of human full-thickness 3D skin tissues that were exposed or not exposed to UVA and UVB for 5 days and topically treated with 0.5% retinol in DMSO, 2% ethyl ascorbic acid in water, 0.1% and 0.2% BG100 extracts in ethanol, 0.5% DMSO, 1% ethanol, or ethanol alone for 6 days. Data are based on duplicate measurements from two skin tissues per each treatment. ** *p* < 0.01; **** *p* < 0.0001, compared to the UV-exposed negative control.

**Figure 5 biomolecules-14-00776-f005:**
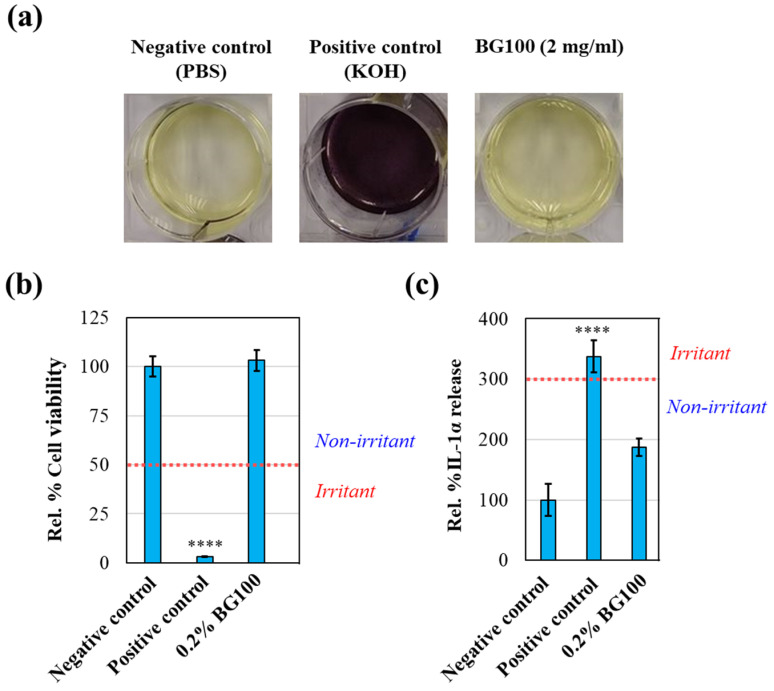
Safety evaluation of BG100 extract on in vitro reconstructed 3D human skin tissues (OECD TG 439). (**a**) MTT interference test performed by adding 1 mg/mL of MTT solution with 0.2% BG100 extract, KOH (positive control), or PBS (negative control). (**b**) Percentage of relative cell viability of the reconstructed human epidermis tissues treated with 0.2% BG100 extract, 5% SDS (positive control), or PBS (negative control) for 1 h and continued incubation for 24 h. (**c**) IL-1α expression in tissue culture medium of reconstructed human epidermis tissues exposed to the same treatments as (**b**). Data are based on duplicate measurements from three skin tissues per each treatment. **** *p* < 0.0001, compared to the negative control.

**Figure 6 biomolecules-14-00776-f006:**
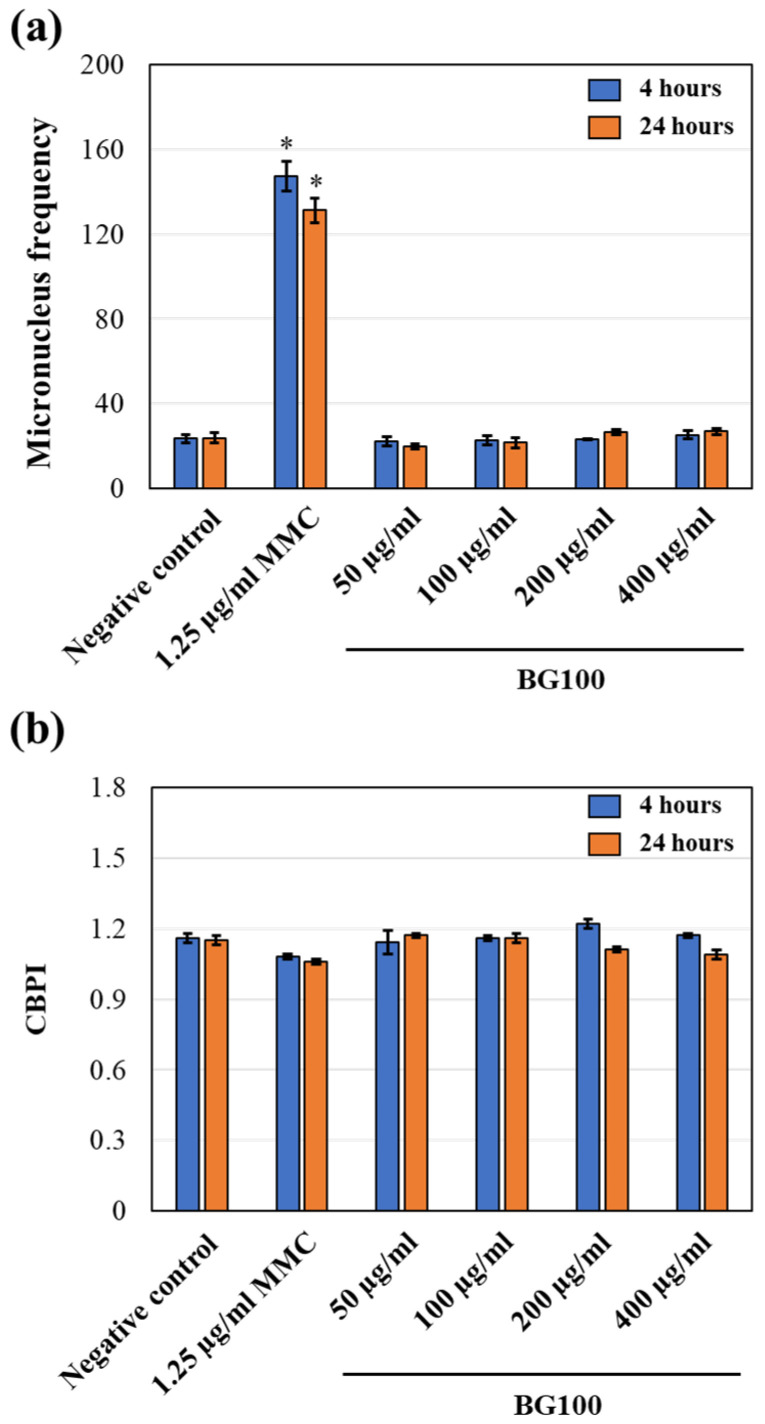
BG100 extract exhibits non-genotoxic effects on V79-4 cells without S9 fraction following 4-h and 24-h treatments. (**a**) Micronucleus (MN) frequency and (**b**) cytokinesis-block proliferation index (CBPI) values were calculated. V79-4 cells were exposed to 50, 100, 200, and 400 µg/mL BG100 extract, 1.25 µg/mL mitomycin C (MMC; positive control), or culture medium (negative control) for 4 or 24 h. Subsequently, cytochalasin B was added and incubation continued for an additional 18–20 h. Data are presented as mean ± SEM from a single independent experiment conducted in triplicate. * *p* < 0.05, compared to the negative control.

**Figure 7 biomolecules-14-00776-f007:**
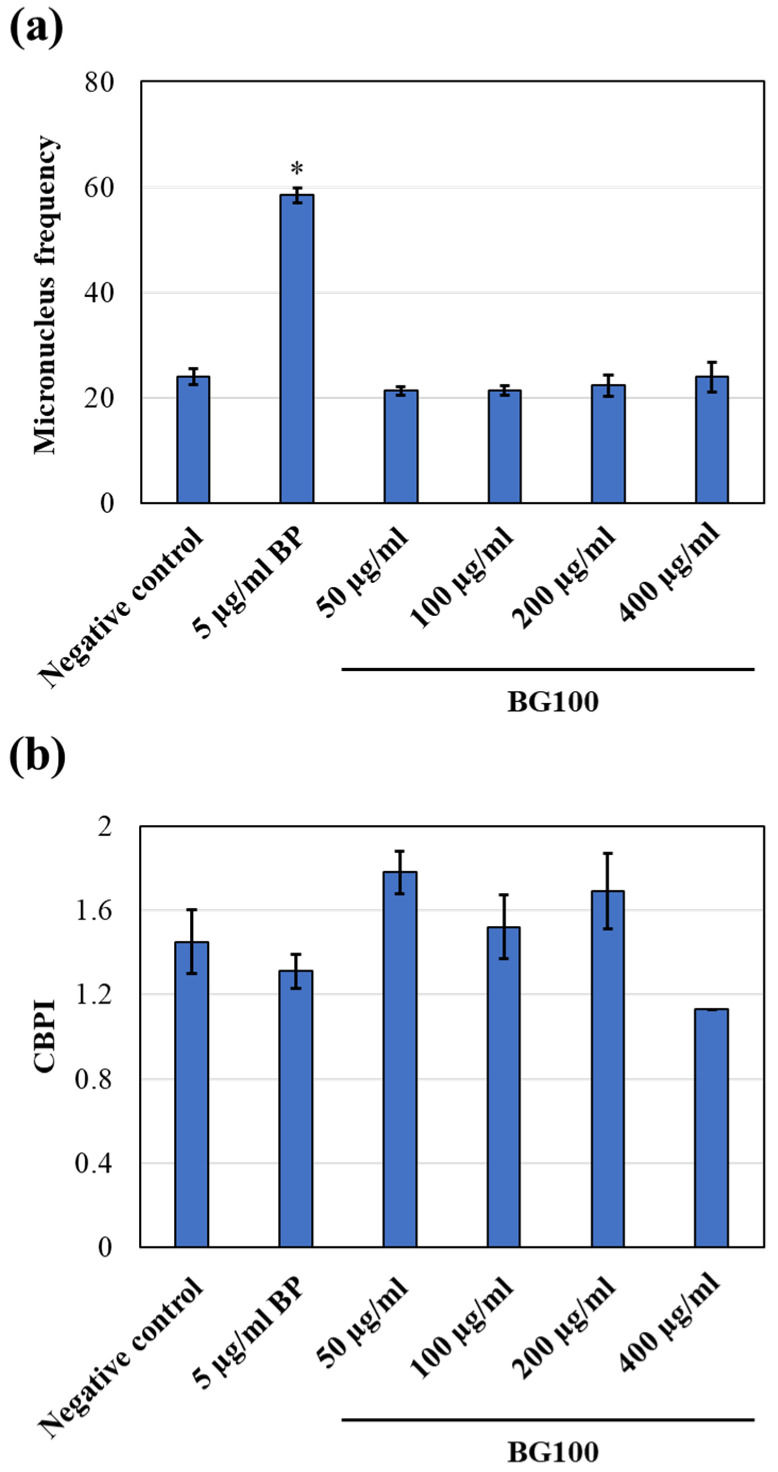
BG100 extract is non-genotoxic to V79-4 cells at concentrations of 50, 100, 200, and 400 µg/mL in the presence of S9 fraction after 4-h treatment. (**a**) MN frequency and (**b**) the CBPI value were calculated. V79-4 cells were treated with 50, 100, 200, and 400 µg/mL BG100 extracts, 5 µg/mL benzo[a]pyrene (BP; positive control), or culture medium (negative control) for 4 h. Cells were treated with S9 fraction for 4 h, then cytochalasin B was added and incubation continued for 18–20 h. MN frequency and CBPI were calculated and reported as Mean ± SEM. Results were performed from one independent experiment with triplicate measurements. * *p* < 0.05, compared to the negative control.

**Figure 8 biomolecules-14-00776-f008:**
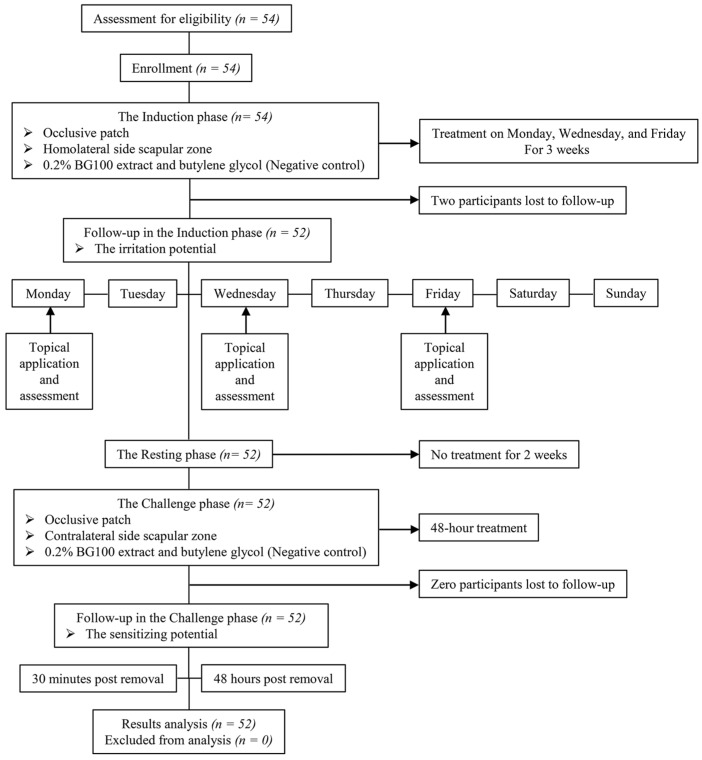
Flow diagram illustrating the clinical study focused on assessing the skin sensitization and irritation potential of BG100 extract.

**Table 1 biomolecules-14-00776-t001:** Clinical criteria regarding the irritation potential (induction phase).

Score	Quotation	Criteria	
		Erythema (E)	Edema (O)
0	Absent	No erythema	No edema
0.5	Very slight	Barely perceptible: pinkish coloration of one part of tested area	Palpable, barely visible
1	Slight	Pinkish coloration of the complete tested area or rather visible on one part of the tested area	Palpable, visible
2	Obvious	Obvious erythema covering the whole tested area	Obvious edema (thickness < 1 mm) with or without blister(s) or vesicle(s)
3	Important	Severe erythema covering all the tested area or obvious erythema diffusing outside the tested area	Severe edema (thickness ≥ 1 mm or diffusing outside the tested area) with or without blister(s) or vesicle(s)

**Table 2 biomolecules-14-00776-t002:** The classification of irritation potential based on the Mean Cumulative Irritation Index (M.C.I.I.).

M.C.I.I.	Class
M.C.I.I. < 0.25	Non-irritating
0.25 ≤ M.C.I.I. < 0.50	Very slightly irritating
0.5 ≤ M.C.I.I. < 1	Slightly irritating
1 ≤ M.C.I.I. < 2	Moderate irritating
M.C.I.I. ≥ 2	Irritating

**Table 3 biomolecules-14-00776-t003:** Clinical criteria regarding the sensitizing potential (challenge phase).

Criterion	ICDRG Quotation	Numeric Score Quotation
No reaction	0	0
Doubtful reaction	?	0.5
Erythema and edema	+	1
Erythema, edema, and vesicles	++	2
Severe reaction with blisters or post-blister ulcerations	+++	3

**Table 4 biomolecules-14-00776-t004:** Sensitizing potential of 0.2% BG100 extract during the challenge phase in healthy human volunteers.

Reading Zone	Criterion	Score	Day 37		Day 39	
			N	%	N	%
Contralateral		T+	1	1.9%	0	0.0%
	No reaction	0	43	82.7%	51	98.1%
	Doubtful reaction	0.5 (?)	8	15.4%	1	1.9%
	Erythema and edema	1 (+)	0	0.0%	0	0.0%
	Erythema, edema, and vesicles	2 (++)	0	0.0%	0	0.0%
	Severe reaction with blisters or post-blister ulcerations	3 (+++)	0	0.0%	0	0.0%

T+: Reactions on the negative control (butylene glycol) and readings were excluded from data analysis, N: Number of subjects, and %: The percentage of subjects.

## Data Availability

Dataset available on request from the authors.

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
