# Peer review of "Skin Rejuvenation Efficacy and Safety Evaluation of *Kaempferia parviflora* Standardized Extract (BG100) in Human 3D Skin Models and Clinical Trial"

_biomolecules, 2024, doi:10.3390/biom14070776_

Round 1
Reviewer 1 Report
Comments and Suggestions for Authors
The research is well organised. The results are interesting and well exposed. A short improvement of conclusions is recommended.
Author Response
Dear Reviewer 1
Thank you very much for your constructive comments and suggestions. Based on your feedback, I have revised the conclusion section of our manuscript to include detailed discussions about future in vivo (pre-clinical) studies for BG100 as an active ingredient in both cosmeceuticals (described in the second paragraph of the conclusion) and nutraceuticals (outlined in the third paragraph). Additionally, I have elaborated on the broader impacts of this project’s success in the final paragraph, emphasizing the potential economic and community benefits.
Our study demonstrates that the standardized BG100 extract, enriched with polymethoxyflavones isolated from Kaempferia parviflora, exhibits significant skin rejuvenation effects. It enhances collagen type I and hyaluronic acid secretion while reducing ROS and DNA damage in photoaged human 3D full-thickness skin models. Furthermore, BG100 extract has proven to be non-genotoxic in vitro, and non-irritating and non-sensitizing to human skin in both in vitro and clinical trials. Consequently, BG100 emerges as a promising and safe anti-aging active ingredient, ideal for integration into cosmeceutical and potentially nutraceutical products.
For the development of cosmeceutical products, we propose conducting future efficacy testing through clinical trials, specifically employing split-face experiments with a sample size of 30 participants. These trials will compare the efficacy of a standard base formulation against the same formulation enriched with BG100, assessing metrics such as hydration, elasticity, skin brightening, and wrinkle reduction. Volunteers selected for these trials will be those regularly exposed to sunlight, aligning clinical outcomes with data previously gathered from 3D skin model studies.
For the potential use of BG100 as an active ingredient in nutraceutical products, ongoing safety and efficacy testing in animal studies is essential. Our collaborative research teams have completed acute and subchronic toxicity studies to ensure safety. Current studies include experiments on mice treated with D-galactose to simulate accelerated aging, aiming to mitigate aging-related damage in various organs and tissues, including skin. These studies are crucial in evaluating improvements in tissue functions, reductions in oxidative stress markers, and decreases in cellular senescence markers, providing essential data for further clinical validation of BG100 for anti-aging and skin rejuvenation.
Ultimately, we envision that the successful development and commercialization of BG100 will position Thailand as one of the leaders in the global market for natural active ingredients. This initiative promises to significantly boost Thailand's economic growth, particularly enhancing the livelihoods and economic stability of its rural communities, showcasing Thailand’s innovation in biotechnology, and promoting sustainable development in its agricultural sector.
I hope this revised version addresses the concerns raised and is now deemed acceptable for publication in Biomolecules. Thank you for the opportunity to refine our work further.
Sincerely,
Tawin Iempridee
Corresponding author

Reviewer 2 Report
Comments and Suggestions for Authors
The Authors reported an article entitled “Skin Rejuvenation Efficacy and Safety Evaluation of Kaempferia parviflora Standardized Extract (BG100) in Human 3D Skin Models and Clinical Trial”.
Their aim is focused on proposing a standardized Kempferia parviflora extract as antiageing ingredient for future applications in cosmetic products and nutraceuticals.
The Authors carefully assess the safety of the studied extract, its quali-quantitative composition by means of HPLC analysis, its biocompatibility with skin tissue, and its efficacy in preserving the photoaged skin tissue.
Furthermore, they evaluated the ROS inhibition exerted by the studied extract in photoaged skin tissue by using 3D models.
Additionally, they estimated the absence of DNA damages, the non-genotoxicity, the non-irritating and non-sensitizing properties of the extract after enrolling healthy volunteers.
The manuscript is well written and organized.
The discussed topic is very interesting, and useful for the Scientific Community.
The experimental design is rigorous and well explained.
For these reasons, I recommend the article for the publication.
Author Response
Dear Reviewer 2,
Thank you for your thorough evaluation of our manuscript and for recommending it for publication. We greatly appreciate your positive feedback on the organization and clarity of the manuscript, as well as your acknowledgment of the rigor of our experimental design and its relevance to the scientific community.
Following suggestions from Reviewer 1 and the editor, we have made revisions to the conclusion section of our manuscript to further elaborate on the potential in vivo (pre-clinical) studies for BG100 as an active ingredient in both cosmeceuticals and nutraceuticals. These revisions aim to enhance the manuscript by outlining future directions and potential impacts more clearly.
We believe that these modifications have strengthened the manuscript and hope that the updated version meets your approval.
Thank you once again for your insightful comments and support.
Best regards,
Tawin Iempridee
Corresponding author
